# Effect of Initial Water Flux on the Performance of Anaerobic Membrane Bioreactor: Constant Flux Mode versus Varying Flux Mode

**DOI:** 10.3390/membranes11030203

**Published:** 2021-03-13

**Authors:** Xiawen Yi, Meng Zhang, Weilong Song, Xinhua Wang

**Affiliations:** Jiangsu Key Laboratory of Anaerobic Biotechnology, School of Environmental and Civil Engineering, Jiangnan University, Wuxi 214122, China; yixiawen_jndx@163.com (X.Y.); AmandaZhangmeng@163.com (M.Z.); swl@jiangnan.edu.cn (W.S.)

**Keywords:** anaerobic membrane bioreactor, initial flux, constant flux, varying flux, membrane fouling, wastewater treatment

## Abstract

Anaerobic membrane bioreactors (AnMBRs) have aroused growing interest in wastewater treatment and energy recovery. However, serious membrane fouling remains a critical hindrance to AnMBRs. Here, a novel membrane fouling mitigation via optimizing initial water flux is proposed, and its feasibility was evaluated by comparing the membrane performance in AnMBRs between constant flux and varying flux modes. Results indicated that, compared with the constant flux mode, varying flux mode significantly prolonged the membrane operating time by mitigating membrane fouling. Through the analyses of fouled membranes under two operating modes, the mechanism of membrane fouling mitigation was revealed as follows: A low water flux was applied in stage 1 which slowed down the interaction between foulants and membrane surface, especially reduced the deposition of proteins on the membrane surface and formed a thin and loose fouling layer. Correspondingly, the interaction between foulants was weakened in the following stage 2 with a high water flux and, subsequently, the foulants absorbed on the membrane surface was further reduced. In addition, flux operating mode had no impact on the contaminant removal in an AnMBR. This study provides a new way of improving membrane performance in AnMBRs via a varying flux operating mode.

## 1. Introduction

Anaerobic membrane bioreactors (AnMBRs) combining anaerobic treatment processes and membrane technology have aroused growing interests in wastewater treatment due to their high contaminant removal, low energy demand, energy recovery, and small footprint [1,2,3,4,5,6,7]. AnMBRs have been successfully developed for treating both high and low strength wastewaters from industries and municipalities, respectively. However, serious membrane fouling remains a critical hindrance to AnMBRs, which results in a decline of water flux, a rise of transmembrane pressure (TMP), an increased cleaning frequency, and a shortened membrane life [8,9,10,11,12,13,14,15,16].

Previous studies have devoted much effort to pursue effective membrane fouling control in AnMBRs. During the last decade, emerging fouling control methods, such as in situ chemical cleaning [17], mechanically assisted aeration scouring [18], electrically assisted fouling mitigation [19], enzymatic and biological degradation of foulants [20], and membrane modifications [21] have been successfully developed in AnMBRs. However, aeration normally consumes much more energy to achieve efficient membrane fouling control; chemical cleaning effect highly depends on the chemicals-foulants interaction; moreover, chemical reagent might accelerate membrane aging and cause microorganism inactivation in bioreactors, and there are still many limits and unknown mechanisms in biological control methods. Fouling mitigation via electric fields has shown great potential for improving membrane permeability very recently while it is still not practically feasible by so far, mainly due to the complex assembly process and consumption of electrodes [22]. Given the above, there is still an urgent need to develop effective methods for membrane fouling control in AnMBRs.

Membrane fouling in AnMBRs can be attributed to cake layer deposition, which is usually the predominant fouling component, and membrane pore clogging [23]. In another words, membrane fouling consists of the undesirable accumulation and deposition of solutes, colloids and microorganisms, and cell debris within and on membranes [24]. Currently, a three-stage fouling process at the constant flux operating mode has been widely recognized including an initial short-term rapid rise in TMP (stage 1), a long-term slight rise in TMP (stage 2), and a TMP jump (stage 3) [25]. The membrane fouling in stage 1 can be attributed to the foulants-membrane surface interaction, while the foulants-foulants interaction has great effects on membrane fouling formation in stage 2, lastly sludge cake layer maturation and compression took place on membrane in stage 3. It has been demonstrated that the initial interaction between foulants and membrane in stage 1 has the greatest impact on the entire membrane fouling process [26]. In other words, weakening the interaction between foulants and membrane, i.e., reducing the membrane fouling in stage 1 may have a positive effect on the entire membrane fouling process. However, a high initial flux could intensify the interaction between foulants and membrane and further aggravated membrane fouling in stage 1 [27]. Therefore, reducing the initial water flux during the operation of AnMBR might be an effective method to control membrane fouling.

In view of above, two flux operating modes, constant initial flux mode with a high initial flux and varying initial flux mode with a low initial flux, were applied in AnMBR to investigate the effect of initial flux on membrane fouling development in this study. As of our best knowledge, study on controlling fouling by optimizing initial water flux has never been addressed before. Overall, the major objective of this study is to evaluate the impact of varying initial water flux on membrane fouling in AnMBR and to explore the feasibility of applying a method of optimizing initial water flux for membrane fouling control. The results shown in current study would be good for better understanding the relationship between initial flux variation and membrane fouling in AnMBR and further developing effective fouling control strategy for AnMBR.

## 2. Materials and Methods

### 2.1. Experimental Setup

The AnMBR used in this study had an effective volume of 7.6 L, which is shown in Figure 1. A flat-sheet microfiltration (MF) membrane module (provided by Tongqin Inc., Shanghai, China) was submerged in the AnMBR, which was made of polyvinylidene fluoride (PVDF) and had an effective area of 0.035 m^2^. Two identical AnMBRs were operated in parallel at the same operating conditions except for the initial water flux for evaluating the impacts of flux variations on the MF membrane performance. The MF membrane flux value was controlled at 6.00 ± 0.36 LMH in the control reactor (named as the constant flux mode), while the other one was controlled at 2.15 ± 0.16 LMH in the first five days (d) of the operation, and then increasing the MF membrane flux value to 6.20 ± 0.31 LMH (named as the varying flux mode). The TMP of the MF membrane was measured via a mercury pressure gauge. The aeration via recirculating part of the produced biogas was applied for strengthening the mixing of the anaerobic biomass in the AnMBR, and the aeration rate was 2 L/min.

The synthetic alcohol wastewater was continuously pumped into both AnMBRs. Its concentrations of ammonia nitrogen (NH_4_^+^-N), chemical oxygen demand (COD), and total phosphorus (TP) were 89.21 ± 5.10, 3284 ± 130, and 18.00 ± 1.49 mg/L, respectively. The seed sludge collecting from Wuxi Xincheng Wastewater Treatment Plant was added into both AnMBRs after cultivating in two fermentation flasks for approximately 50 days at the temperature of 35 ± 1 °C. The initial sludge concentrations were 6.15 and 4.95 g/L for the mixed liquor suspended solids (MLSS) and mixed liquor volatile suspended solids (MLVSS), respectively, in both AnMBRs. Throughout the operation, the two reactors were operated at the temperature of 35 ± 1°C. The SRT was set as 100 days for both reactors.

### 2.2. Analytical Methods

Measurements of NH_4_^+^-N, COD, and TP were determined in the influent, effluent and sludge supernatant according to the Standard method [28]. A field emission scanning electron microscope (FESEM, S4800, Hitachi, Japan) and an energy diffusive X-ray (EDX, Quantax, Bruker, Germany) were applied for characterizing the morphology and element compositions of the MF membranes, respectively. The distributions of biofoulants, including microorganisms, proteins, α-d-glucopyranose and β-d-glucopyranose polysaccharides on the MF membrane were analyzed via a confocal laser scanning microscope (CLSM, LSM 710, Carl Zeiss, Germany). Before observing by the CLSM, these biofoulants were firstly stained by SYTOTM 63, Fluorescein isothiocyanate (FITC), Concanavalin A (ConA) and Calcofluor white (CW), respectively. The specific information on the CLSM analyses can be found in previous literature [29]. The ZEN software was used to obtain the three-dimensional CLSM images, and the Auto PHLIP-ML software (version 1.0) was applied for calculating the biovolume of the biofoulants.

## 3. Results and Discussion

### 3.1. Performances at Different Operating Modes

The treatment performances under two operating modes were shown in Table 1. Regardless of the initial water flux, both AnMBRs had a high COD removal efficiency of about 94.0% for both operating modes. The high COD removal rate was mainly owing to the rejection of MF membrane and the bio-degradation of microorganisms. Specifically, the bio-degradation process made the predominant contribution to COD removal (indicated by the large reduction of COD in supernatant to that in the influent). However, the removals of NH_4_^+^-N and TP were not as good as COD in both operating modes. It might be owing to the anaerobic condition and low membrane retention effect in the AnMBR. In addition, the similar COD and NH_4_^+^-N treatment performance were achieved under two operating modes. However, the TP concentration in the effluent of the varying flux operating mode was higher than that of the constant flux operating mode. This may be due to the accumulation of phosphorus in the reactor and the gradual decrease in the use of phosphorus by microorganisms during long-term operation.

### 3.2. The Development of TMP and Cumulative Water Production at Different Operating Modes

Variation of TMP and the cumulate production capacity in both AnMBRs are illustrated in Figure 2. As shown in Figure 2a, the variation of TMP in the AnMBR with the constant flux mode consisted of three main stages, which was consistent with the three stages fouling theory. In the first stage (days 1–2), an initial short-term rapid rise in TMP from 1.00 kPa to 3.25 kPa was observed. After that, the TMP proceeded to a long-term slow rise (days 3–24) from 3.25 kPa to 11.18 kPa. The last stage (days 25–31) had a dramatic TMP jump. Based on previous reports [23,30], the initial TMP rise in stage 1 was attributed to the deposition of foulants on the MF membrane surface, which was dominated by the interaction between foulants and the MF membrane surface; after that, the TMP rise slows down in stage 2 because the governing force has turned to the interaction between foulants and foulants when a fouling layer was formed on the membrane surface; in stage 3, the TMP jump was mainly owing to the formation of a compacted cake layer on the membrane surface. However, the variation of TMP in the AnMBR under the varying flux mode only presented the last two stages, i.e., a long-term slow rise stage and a sudden jump stage (shown in Figure 2b). According to the three stages fouling theory, this phenomenon might be due to the lower initial water flux of MF membrane which might reduce the deposition of foulants on the membrane surface in stage 1 and, thus, alleviating membrane fouling. Additionally, it was calculated that the average TMP increasing rates of constant flux mode and varying flux mode were 0.88 and 0.34 kPa/d, respectively, which means that the operation time under the varying flux mode was much longer than that of the constant flux mode. Therefore, there was much more cumulative water permeating volume in the varying flux mode in spite of the low initial membrane flux. Thus, it can be speculated that reducing the initial flux was a feasible way to improve the performance of AnMBRs.

### 3.3. Fouling Characteristics at Different Operating Modes

At the end of experiments, the fouled MF membranes from both AnMBRs under different operating modes were collected for fouling characteristics analyses. As shown in Figure 3a,b, all fouled MF membranes were covered with a fouling layer. Moreover, crystals were also observed on the surface of fouled MF membranes. It can be observed from Figure 3c,d, there was no significant differences between the fouled MF membranes in term of element composition. The presence of C, N, Ca, Mg, and P indicated that there were both biofouling and inorganic scaling on the fouled MF membranes. In addition, the signal of C and N peak was much larger than those of the inorganic elements, such as Ca, Mg, and P, implying that biofouling had a dominant role in membrane fouling of the AnMBR. And the higher C peak of constant flux operating mode might be due to the fact that there were more biofoulants on the membrane surface of constant flux operating mode than varying flux operating mode.

Based on the major contribution of biofouling to the membrane fouling in the AnMBR, the biofoulants consisting of microorganisms, polysaccharides and proteins on the MF membranes under different operating modes were further analyzed by the CLSM coupled with multiple fluorescence labeling. The distributions of biofoulants are shown in Figure 4 and the color of red, green, cyan and blue indicated microorganisms, proteins, α-D-glucopyranose, and β-D-glucopyranose polysaccharides, respectively. As shown in Figure 4, the thickness of fouling layers formed on the membrane surface at the constant flux and varying flux modes was 94 and 42 μm, respectively, and the area of biofoulants was much darker at constant flux mode than that at varying flux mode. The reduced thickness of fouling layer and the much lighter fouling area indicated that the biofouling was mitigated in the AnMBR under the varying flux operating mode. In addition, the biovolume of biofoulants was calculated for analyzing their constituents and contents on the MF membrane surfaces. As shown in Table 2, compared to the constant flux mode, there were less proteins, polysaccharides and microorganisms discovered on the surface of membrane under the varying flux mode, especially the content of proteins which reduced from 8.2 to 2.37 μm^3^/μm^2^. It implied that the reduction of proteins might be the main reason for the alleviation of membrane fouling [31,32] because proteins were the most typical foulants causing irreversible membrane fouling [33,34]. Therefore, owing to the reduction of the deposited biofoulants and the fouling layer thickness by lowering initial flux, MF membrane fouling in the AnMBR was effectively alleviated under the varying flux mode and its operating time was prolonged as well.

### 3.4. Implications

In view of above results, the high initial water flux enhanced the deposition of foulants on the membrane surface in Stage 1 because of the stronger driving force. Based on this fact, we proposed that the operating performance of MF membrane can be improved via reducing the filtration driving force in stage 1. This idea can be achieved by changing the operating mode of the membrane flux. Specifically, the MF membrane was controlled at a low initial flux in stage 1, subsequently restored the flux to a normal level in stage 2.

To sum up, applying the varying flux operation mode was a feasible method to improve the operating performance of MF membrane in the AnMBR. It was owing to a low driving force in stage 1, which reduced the interaction between pollutants and MF membrane and, thus, mitigating the deposition of foulants on the membrane surface. Consequently, the foulant-foulant interaction and fouling development in stage 2 were further mitigated due to the thin and loose fouling layer formed in stage 1. As a result, longer operating time of MF membrane and the consequent higher cumulative permeate production can be achieved under varying flux mode.

## 4. Conclusions

In this study, the performances of the constant and varying flux operating mode were investigated. Compared with the constant flux operating mode, the running time of the varying flux operating mode was much longer. And the cumulative water production of the varying flux operating mode was more than 300 L in 75 d, which was more than twice of constant flux operating mode. Additionally, there were less foulants, especially proteins, depositing on the membrane surface under the varying flux mode and the total biovolume of the biofoulants was 19.44 ± 0.80 μm^3^/μm^2^. In summary, applying varying flux operating mode was a feasible way to improve the operating performance of MF membrane in AnMBR.

## Figures and Tables

**Figure 1 membranes-11-00203-f001:**
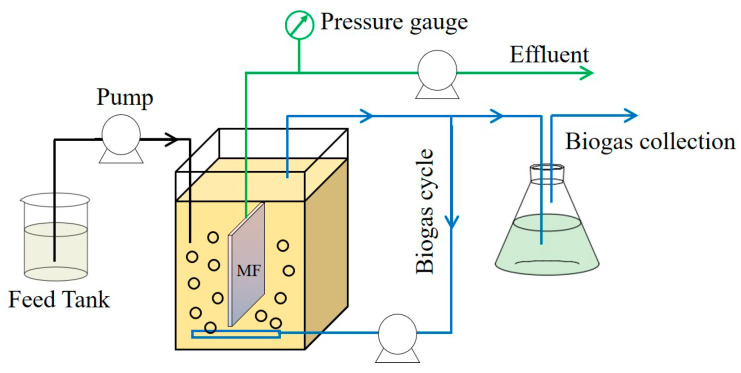
Schematic diagram of the anaerobic membrane bioreactor (AnMBR) system.

**Figure 2 membranes-11-00203-f002:**
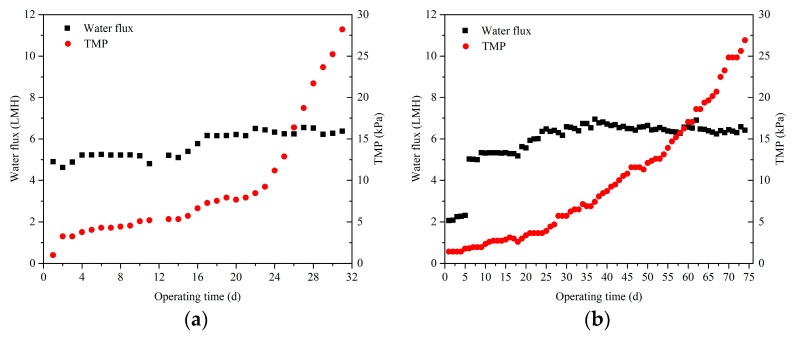
Variations of transmembrane pressure (TMP) (**a**,**b**) and cumulate production capacity (**c**) at different operating modes of water flux. (**a**) Constant flux mode; (**b**) varying flux mode.

**Figure 3 membranes-11-00203-f003:**
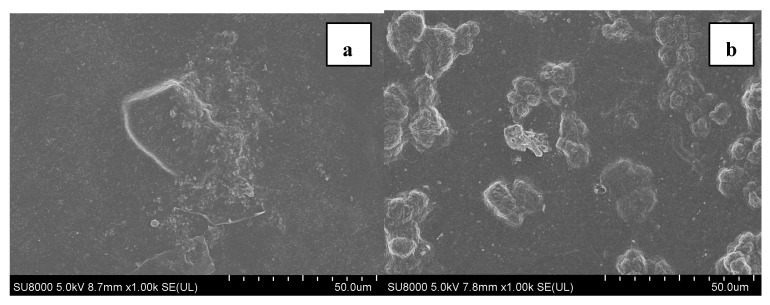
Surface images of the fouled microfiltration (MF) membranes: SEM images and EDX spectrums at different operating modes of constant flux (**a**,**c**) and varying flux (**b**,**d**).

**Figure 4 membranes-11-00203-f004:**
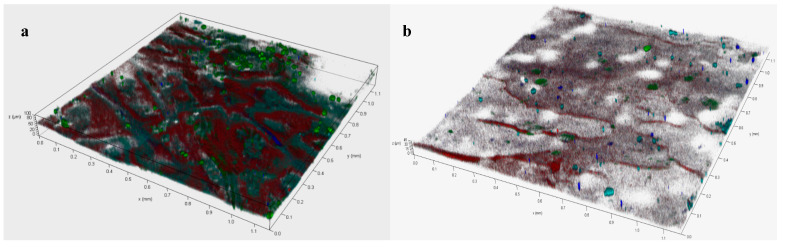
Images of the fouled MF membranes obtained from confocal laser scanning microscope (CLSM) at different operating modes of constant flux (**a**) and varying flux (**b**).

**Table 1 membranes-11-00203-t001:** COD, TP, and NH_4_^+^-N concentrations in MF permeates and the removal performances at different operating modes.

Items	Performance	Constant Flux Mode	Varying Flux Mode
COD	MF permeate concentration (mg/L)	193.55 ± 19.35	196.64 ± 16.26
	Removal rate (%)	94.05 ± 0.95	94.06 ± 2.66
NH_4_^+^-N	MF permeate concentration (mg/L)	67.35 ± 1.53	62.10 ± 4.58
	Removal rate (%)	25.62 ± 6.70	29.77 ± 3.81
TP	MF permeate concentration (mg/L)	6.25 ± 1.48	14.18 ± 1.78
	Removal rate (%)	4.38 ± 1.93	18.36 ± 1.13

**Table 2 membranes-11-00203-t002:** Biovolume of the biofoulants on the fouled MF membranes at different operating modes calculated by PHLIP.

Operating Mode	Total Cells (μm^3^/μm^2^)	Proteins (μm^3^/μm^2^)	α-d-glucopyranose (μm^3^/μm^2^)	β-d-glucopyranose (μm^3^/μm^2^)
Constant flux	7.71 ± 0.40	8.20 ± 0.25	8.29 ± 0.38	7.72 ± 0.49
Varying flux	6.64 ± 0.36	2.37 ± 0.29	5.60 ± 0.28	4.26 ± 0.20

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
