# Peer review of "Effect of Initial Water Flux on the Performance of Anaerobic Membrane Bioreactor: Constant Flux Mode versus Varying Flux Mode"

_membranes, 2021, doi:10.3390/membranes11030203_

Round 1
Reviewer 1 Report
Check and revise the followed for your better article.
- Line 20: "the interaction between foulants and foulants" should be revised to "the interaction between foulants".
- Line 45: A reference related with "Fouling mitigation ~ process and consumption of electrodes." should be added.
- Line 81: Add the MF module in your study is hollow fiber or flat sheet.
- Line 121: The treatment performance of TP was very different under two operating modes in Table 1.
- Line 167: The C peak of constant flux is much higher than that of varying flux. So explain this result.
Author Response
We thank the reviewer #1 for his/her comments of our manuscript. The detailed response and revisions to all comments raised by the reviewer are summarized in the following section. 1. Line 20: "the interaction between foulants and foulants" should be revised to "the interaction between foulants". Response: Thank the reviewer for his/her serious reading of our manuscript. The mistake has been corrected in the revised manuscript. 2. Line 45: A reference related with "Fouling mitigation ~ process and consumption of electrodes." should be added. Response: Thank the reviewer for his/her suggestion. The reference has been added in the revised manuscript. Specific revisions in the revised manuscript: (1) Line 49: A new reference was supplemented in the revised manuscript as follows: Li, C.Y.; Guo, X.Y.; Wang, X.; Fan, S.G.; Zhou, Q.X.; Shao, H.W.; Hu, W.L.; Li, C.H.; Tong, L.; Kumar, R.R.; Huang, J.H. Membrane fouling mitigation by coupling applied electric field in membrane system: Configuration, mechanism and performance. Electrochimica Acta 2018, 287, 124-134, doi: 10.1016/j.electacta.2018.06.150. 3. Line 81: Add the MF module in your study is hollow fiber or flat sheet. Response: We appreciated reviewer’s suggestion. The type of MF module has been added in the manuscript. Specific revisions in the revised manuscript: (1) Line 89: The type of MF module was supplemented in the revised manuscript as follow: A flat-sheet MF membrane module (provided by Tongqin Inc., China) was submerged in the AnMBR, which was made of polyvinylidene fluoride (PVDF) and had an effective area of 0.035 m2. 4. Line 121: The treatment performance of TP was very different under two operating modes in Table 1. Response: Thank the reviewer for his/her serious reading of our manuscript. The difference in the TP treatment performance may be caused by the different running time. As shown in the manuscript, the running time of the varying flux mode was more than twice of the constant flux mode. On the one hand, long-term operation resulted in the accumulation of phosphorus in the reactor and the retention performance of MF membrane was limited. On the other hand, the microorganisms in the reactor under the varying flux operating mode may have passed the logarithmic phase after a long operation and the need for phosphorus may be reduced. Therefore, the TP concentration in the effluent under varying flux operating mode was higher than that under constant flux operating mode. Specific revisions in the revised manuscript: (1) Line 138-143: The explanations on different TP treatment performance were supplemented in the revised manuscript as follow: In addition, the similar COD and NH4+-N treatment performance were achieved under two operating modes. However, the TP concentration in the effluent of varying flux operating mode was higher than that of constant flux operating mode. This may be due to the accumulation of phosphorus in the reactor and the gradual decrease in the use of phosphorus by microorganisms during long-term operation. 5. Line 167: The C peak of constant flux is much higher than that of varying flux. So explain this result. Response: The higher C peak of constant flux operating mode might be due to the fact that there were more biofoulants on the membrane surface of constant flux operating mode than varying flux operating mode. And the explanation of the result has been supplemented in the revised manuscript. Specific revisions in the revised manuscript: (1) Line 186-188: The explanation of higher C peak of constant flux operating mode was supplemented in the revised manuscript as follow: And the higher C peak of constant flux operating mode might be due to the fact that there were more biofoulants on the membrane surface of constant flux operating mode than varying flux operating mode.
Reviewer 2 Report
This is a well-written, well-organized and well-illustrated paper. The results of this paper are particularly interesting.
I recommend this paper to be accepted .
Author Response
Thank the reviewer #2 very much for his/her positive comments.Reviewer 3 Report
The manuscript "Effect of initial water flux on the performance of anaerobic membrane bioreactor: constant flux mode versus varying flux mode" described an anaerobic membrane bioreactor (AnMBR) as a wastewater treatment and energy recovery.
In this work, authors demonstrated that the fouling development in AnMBR was largely determined by the initial water flux. The results are interesting but in my opinion, the conducted research and especially the discussion are not of a apropriate level to be published in the Membranes. But before next submission authors should carefully consider the following suggestions:
1. The introduction should be more specify and in particular, authors should emphasized why took up this subject.
2. Figure 4 is difficult to read.
3. The conclusion should be refined, because it is too lack.
4. Generally, the publication looks more like a preliminary study rather than a regular research paper.
Author Response
The comments given by reviewer #3 is a great help for us to improve our manuscript. The major and specific comments raised by the reviewer together with our modifications are listed as follows.
- The introduction should be more specify and in particular, authors should emphasize why took up this subject.
Response: We appreciated reviewer’s suggestion. The structure of the introduction has been modified and the basis of the subject has also been improved.
Specific revisions in the revised manuscript:
(1) Line 51-78: The adjustment and modification of introduction were shown in the revised manuscript as follow:
Membrane fouling in AnMBR can be attributed to cake layer deposition, which is usually the predominant fouling component, and membrane pore clogging [23]. In another word, membrane fouling consists of the undesirable accumulation and deposition of solutes, colloids and microorganisms, and cell debris within and on membranes [24]. Currently, a three-stage membrane fouling process at the constant flux operating mode has been widely recognized, including an initial short-term rapid rise in TMP (Stage 1), a long-term slight rise in TMP (Stage 2) and a TMP jump (Stage 3) [25]. The membrane fouling in Stage 1 can be attributed to the foulants-membrane surface interaction, while the foulants-foulants interaction has great effects on membrane fouling formation in Stage 2, lastly sludge cake layer maturation and compression took place on membrane in Stage 3. It has been demonstrated that the initial interaction between foulants and membrane in Stage 1 has the greatest impact on the entire membrane fouling process [26]. In other words, weakening the interaction between foulants and membrane, i.e., reducing the membrane fouling in Stage 1 may have a positive effect on the entire membrane fouling process. However, a high initial flux could intensify the interaction between foulants and membrane and further aggravated membrane fouling in Stage 1 [27]. Therefore, reducing the initial water flux during the operation of AnMBR might be an effective method to control membrane fouling.
In view of above, two flux operating modes, constant initial flux mode with a high initial flux and varying initial flux mode with a low initial flux, were applied in AnMBR to investigate the effect of initial flux on membrane fouling development in this study. As of our best knowledge, study on controlling fouling by optimizing initial water flux has never been addressed before. Overall, the major objective of this study is to evaluate the impact of varying initial water flux on membrane fouling in AnMBR and to explore the feasibility of applying a method of optimizing initial water flux for membrane fouling control. The results shown in current study would be good for better understanding the relationship between initial flux variation and membrane fouling in AnMBR and further developing effective fouling control strategy for AnMBR.
- Figure 4 is difficult to read.
Response: Thank the reviewer for his/her suggestion. The distributions of biofoulants including microorganisms, proteins, α-D-glucopyranose and β-D-glucopyranose polysaccharides on the MF membrane are shown in Figure 4. Each color represents a type of biofoulant and the color of red, green, cyan and blue represents the distributions of microorganisms, proteins, α-D-glucopyranose and β-D-glucopyranose polysaccharides, respectively. Besides, more darker area indicated more biofoulants. The figure has been re-described in the manuscript for reader to understand.
Specific revisions in the revised manuscript:
(1) Line 219-226: The description of Figure 4 was improved in the revised manuscript as follow:
The distributions of biofoulants are shown in Figure 4 and the color of red, green, cyan and blue indicated microorganisms, proteins, α-D-glucopyranose and β-D-glucopyranose polysaccharides, respectively. As shown in Figure 4, the thickness of fouling layers formed on the membrane surface at the constant flux and varying flux modes was 94 and 42 μm respectively and the area of biofoulants was much darker at constant flux mode than that at varying flux mode. The reduced thickness of fouling layer and the much lighter fouling area indicated that the biofouling was mitigated in the AnMBR under the varying flux operating mode.
- The conclusion should be refined, because it is too lack.
Response: Thank the reviewer for his/her serious reading of our manuscript. The main conclusion has been improved in the revised manuscript.
Specific revisions in the revised manuscript:
(1) Line 269-285: The conclusion was improved in the revised manuscript as follows:
In this study, the performances of the constant and varying flux operating mode were investigated. Compared with the constant flux operating mode, the running time of the varying flux operating mode was much longer. And the cumulative water production of the varying flux operating mode was more than 300 L in 75 d, which was more than twice of constant flux operating mode. Besides, there were less foulants, especially proteins, depositing on the membrane surface under the varying flux mode and the total biovolume of the biofoulants was 19.44 ± 0.80 μm3/μm2. In summary, applying varying flux operating mode was a feasible way to improve the operating performance of MF membrane in AnMBR.
- Generally, the publication looks more like a preliminary study rather than a regular research paper.
Response: We are very grateful for the reviewer’s comments. We have carefully checked the whole manuscript and some discussions and analysis of the experimental results also have been supplemented to make the manuscript more informative.

Round 2
Reviewer 3 Report
I recommend this work for publication in Membranes as it is.